# Filtering Discomforting Recommendations with Large Language Models

## Abstract

Personalized algorithms can inadvertently expose users to discomforting recommendations, potentially triggering negative consequences. The subjectivity of discomfort and the black-box nature of these algorithms make it challenging to effectively identify and filter such content. To address this, we first conducted a formative study to understand users' practices and expectations regarding discomforting recommendation filtering. Then, we designed a Large Language Model (LLM)-based tool named DiscomfortFilter, which constructs an editable preference profile for a user and helps the user express filtering needs through conversation to mask discomforting preferences within the profile. Based on the edited profile, DiscomfortFilter facilitates the discomforting recommendations filtering in a plug-and-play manner, maintaining flexibility and transparency. The constructed preference profile improves LLM reasoning and simplifies user alignment, enabling a 3.8B open-source LLM to rival top commercial models in an offline proxy task. A one-week user study with 24 participants demonstrated the effectiveness of DiscomfortFilter, while also highlighting its potential impact on platform recommendation outcomes. We conclude by discussing the ongoing challenges, highlighting its relevance to broader research, assessing stakeholder impact, and outlining future research directions.

## CCS Concepts

• **Information systems** → **Personalization**.

## Keywords

discomforting recommendation filtering, large language model

**ACM Reference Format:**
Anonymous Author(s). 2018. Filtering Discomforting Recommendations with Large Language Models. In *Proceedings of Make sure to enter the correct conference title from your rights confirmation emai (Conference acronym 'XX).* ACM, New York, NY, USA, 14 pages. https://doi.org/XXXXXXX.XXXXXXX

## 1 Introduction

Personalized algorithms, which analyze user preferences to deliver tailored content and thereby support human decision-making, are indispensable across web platforms [45, 70]. While these algorithms are designed to enhance user experience, they can inadvertently

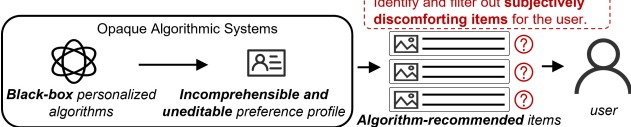

Figure 1: Problem formulation.

expose users to discomforting recommendations [63]. For example, if a user searches for sensitive topics like health issues, the algorithm might suggest related health products, which could be perceived as a breach of privacy and cause unease [81]. Similarly, when a user is experiencing emotional distress, such as after a breakup, algorithms lacking contextual awareness might recommend content that evokes painful memories, potentially worsening the user's emotional state [66]. Such recommendations may not only fail to engage users but also lead to negative emotional consequences, such as anxiety, unease, or distress [63]. The perception of discomfort is highly subjective, meaning that content one user finds enjoyable may be discomforting to another [63, 65, 79, 80]. This subjectivity underscores the urgent need for a more nuanced approach to discomforting content identification that aligns more closely with individual user experiences [30, 67, 74].

In this paper, we aim to design a tool that helps users filter out discomforting recommendations. This task has two key challenges: (1) users' perceptions of discomfort are highly subjective, and (2) the algorithms recommending such content operate as black-box systems. As illustrated in Figure 1, we formalize the problem as follows: black-box personalized algorithms recommend items to a user based on the inferred preference profile (often implicit in the embeddings), and our objective is to identify and filter out subjectively discomforting items for the user. Due to the opacity of algorithmic systems, the preference profile is both incomprehensible and uneditable, making it challenging for the user to influence the algorithm's decisions.

To inform our design process, we conducted a formative study to gain insights into the current landscape and user expectations (Section 3). Initially, we identified several key factors contributing to discomforting recommendations, including deviations in user behavior, biases in algorithmic modeling, and conflicting interests among stakeholders. We then examined the limitations of current feedback mechanisms, emphasizing shortcomings such as insufficient personalization, inflexibility, and a lack of transparency. Based on these findings, we established four design goals to guide the design of our tool: support conversational configuration, provide preference explanations, provide feedback channels, and operate in a plug-and-play manner.

Given the natural language understanding, reasoning, and generation capabilities demonstrated by LLMs [49, 93], we propose that LLM provide a promising solution for achieving these design goals. To this end, we designed an LLM-based tool named DiscomfortFilter,

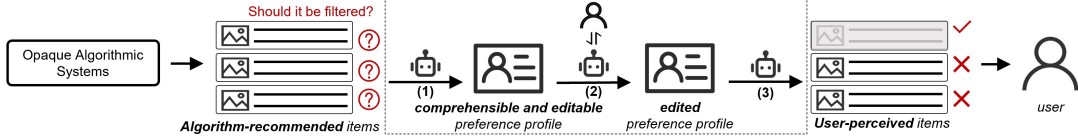

Figure 2: The workflow of DiscomfortFilter.

specifically aimed at helping users filter out discomforting recommendations (Section 4). Figure 2 illustrates the workflow of DiscomfortFilter: (1) DiscomfortFilter identifies algorithm-recommended items based on the user's personalized perceptions, integrates the user's pairwise preferences, and ranks them to construct a comprehensible and editable preference profile tailored for the user; (2) Through a guided conversation, DiscomfortFilter assists the user in expressing personalized filtering needs, and then masks the discomforting preferences with in the profile; (3) DiscomfortFilter filters out discomforting recommendations based on the edited preference profile in a plug-and-play manner, ensuring that user-perceived items no longer includes discomforting elements. Additionally, DiscomfortFilter provides the user with access to filtering logs recorded during step (3), assisting the user in refining filtering needs in a manner similar to step (2). Overall, DiscomfortFilter empowers the user to actively influence the decisions made by personalized algorithms, enhancing control over the algorithms.

We validated the efficacy of the constructed preference profile using an offline proxy task. The findings suggest that it enhances the reasoning process of LLMs, markedly decreases the challenge of aligning them with users, and allows a 3.8B open-source LLM to rival top commercial models. Additionally, we conducted a one-week user study on Zhihu (a platform similar to Quora), China's largest Q&A community, with 24 participants (Section 5). The results demonstrate that our design goals effectively help users express their filtering needs and filter out discomforting recommendations, with DiscomfortFilter successfully achieving these goals. We also analyzed how DiscomfortFilter impacts platform recommendation outcomes by influencing the exposure of discomforting items. Finally, we conducted an in-depth discussion (Section 6), covering: challenges of filtering discomforting recommendations with LLMs, relevance to research topics in recommender systems, potential impact on platforms, and limitations and future work.

The key contributions of this work are outlined below:

- We conducted a formative study with 15 participants to examine the current status and user expectations regarding the filtering of discomforting recommendations.
- We designed an LLM-based tool named DiscomfortFilter to assist users in filtering out discomforting recommendations, which is the first attempt to leverage LLMs in this important task, to the best of our knowledge.
- We evaluated DiscomfortFilter through an offline proxy experiment and a user study and the results showed that DiscomfortFilter can effectively help users express their filtering needs and filter out discomforting recommendations.
- We discussed the challenges and opportunities of using LLMs for filtering discomforting recommendations and shed light on its broader implications.

## 2 Related Work

Personalized algorithms primarily derive user preferences from behavioral data, leading to incomplete user modeling and discomforting recommendations [70]. In Appendix E.1, we provide a detailed review of studies illustrating this phenomenon across various scenarios, including privacy invasion [81], lack of contextual understanding [66], popularity bias [20], and information bubbles [71], along with their negative outcomes [80]. Although previous research has identified scenarios and causes of discomfort, **few studies have focused on designing systems to identify and filter these recommendations, which our work aims to address**.

In Appendix E.2, we provide a detailed review of two potential solutions for filtering discomforting recommendations: interactive recommendation systems [23, 31, 52] and content moderation systems [13, 29, 36]. Both allow users to modify recommendations to mitigate discomfort. However, interactive systems often struggle to scale in opaque environments due to their reliance on specific algorithm designs, while moderation systems focus on objectively harmful content, which may be less effective for addressing subjective discomfort. Thus, **the subjectivity of discomfort perception and the opacity of algorithms present notable challenges**.

In Appendix E.3, we provide a detailed review of studies on LLMs as personal assistants [49], covering commercial applications [59], frameworks for human-centered recommender systems [76], and designs tailored for special populations [16]. The impressive capabilities of LLMs, especially in handling personal data and services, underscore their potential for effectively filtering discomforting recommendations. **To our knowledge, this is the first exploration of using LLMs for this specific purpose.**

## 3 Formative Study

We conducted semi-structured interviews and participatory design sessions with **15** participants, each lasting approximately **one hour**. Additional details about the process are provided in Appendix A.

### 3.1 Findings from Semi-Structured Interviews

During the semi-structured interviews, we explored participants' experiences with discomforting recommendations and the issues they faced when using the "Not Interested" button[1] for feedback. Our analysis identified two key findings from their responses.

**F1: Users may encounter discomforting recommendations for three reasons. (1) User behavior deviation.** Curiosity-driven search behavior and clickbait-induced clicks may fail to reflect a user's true long-term interests, leading inaccurate user preference modeling. For example, P03 said "*Out of curiosity, I once searched*

---

[1]Figure 9 shows the button interfaces of the four platforms mentioned most frequently.

*for adult products, and now they keep showing up in my recommendations—so embarrassing.*" **(2) Algorithmic modeling bias.** Personalized algorithms cannot fully capture the nuanced interests[2] and contexts of users. For example, P06 said "*Getting horror content at night is awful, even if I watch it during the day.*" **(3) Conflicting interests.** For instance, platforms may promote content designed to boost user engagement, even if it may cause discomfort. Seven participants mentioned scenarios where this was the case.

**F2: Platforms' "Not Interested" button faces three major limitations that reduce user engagement. (1) Lack of personalization.** Thirteen participants found the options too vague, making it difficult for them to articulate their specific reasons and potentially leading to the unintended exclusion of content they might otherwise enjoy. **(2) Lack of flexibility.** Eleven participants expressed concern that content they temporarily wish to hide might be permanently removed from their feed. **(3) Lack of transparency.** Twelve participants expressed dissatisfaction with the uncertainty about whether their feedback was being processed effectively.

## 3.2 Design Goals Established through Participatory Design

In response to the issues mentioned above, during the participatory design process, we further discussed participants' specific expectations for the tool and summarized the following four design goals.

**G1: Support conversational configuration.** Thirteen participants indicated that they prefer expressing their filtering needs using natural language because it "*isn't limited by predefined options*" (P11) and "*allows for more accurate and personalized expression*" (P04). Additionally, ten participants expressed a desire to communicate their filtering needs through conversation with the tool, as it feels "*more natural*" (P09). By supporting conversational configuration, the issue of "lack of personalization" is addressed.

**G2: Provide preference explanations.** As the input shifts from predefined options to open conversations, 10 participants reported difficulty in proactively articulating filtering needs. However, all participants agreed that understanding the preferences reflected in platform recommendations and their own behavior would encourage them to express these needs. For example, P05 said, "*I can review it and then provide targeted feedback on any inaccuracies.*"

**G3: Provide feedback channels.** All participants expressed the need for the tool to exhibit transparency and be contestable. Being informed about the filtered content and the corresponding reasons can "*enhance trust in the tool*" (P13), while allowing corrections to the tool's behavior helps "*refine filtering needs*" (P12). By providing feedback channels, the issue of "lack of transparency" is addressed.

**G4: Operate in a plug-and-play manner.** The plug-and-play approach means that the tool operates independently of specific personalized algorithms and directly affects the outputs of these algorithms. Three key factors support this: (1) Participants recognized that their filtering needs are dynamic, as "*discomforting content varies by state*" (P06); (2) Nine participants highlighted that the tool should be user-managed, enabling it to "*work across platforms*" (P14); (3) Participants were more concerned with the discomfort

---

[2]This stems from the fact that collaborative filtering algorithms mainly retain low-frequency information during model training [75].

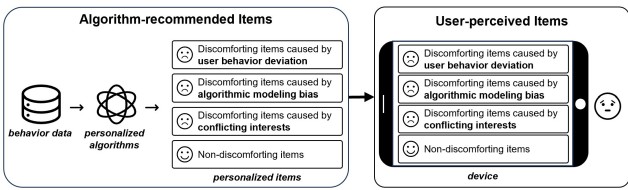

**Figure 3: The process of presenting items to a user before introducing DiscomfortFilter.**

caused by personalized algorithmic outputs than with understanding the algorithms themselves. By operating in a plug-and-play manner, the issue of "lack of flexibility" is addressed.

## 4 DiscomfortFilter

We designed and implemented an LLM-based tool, DiscomfortFilter, which meets the four design goals established in the formative study and aims to assist users in filtering discomforting recommendations. The workflow of DiscomfortFilter has already been illustrated in Figure 2, and this section will provide a detailed introduction.

### 4.1 Overview

Figure 3 illustrates how personalized algorithms present items to a user before the introduction of DiscomfortFilter. These algorithms analyze user behavior to recommend items, which may include both discomforting and non-discomforting items. The personalized items is then displayed on the user's device for **passive consumption**.

The upper part of Figure 4 illustrates how personalized algorithms present items to the user after the introduction of DiscomfortFilter. By integrating a *Content Filter Module* into the original items presentation flow, DiscomfortFilter removes discomforting recommendations, ensuring that only non-discomforting items are ultimately displayed to the user. The identification of discomfort is based on user-configured *filtering rules*, giving the user **an ability of control** over the process.

The filtering rules, existing in natural language form, are crucial for the operation of DiscomfortFilter. The lower part of Figure 4 illustrates how the user configures these rules. The user can manage them directly (green arrow) or utilize the *Filtering Needs Discovery Module* for conversational rule configuration (blue arrow). This conversational agent employs two strategies to help the user identify filtering needs: the first strategy relies on the preference profile constructed by the *Preference Profile Construction Module*, while the second strategy relies on filtering records from the Content Filter Module. The *Candidate Rule Generation Module* analyzes the user's filtering needs from the conversations and translates them into management actions for the filtering rules, which the user can then edit and confirm.

### 4.2 Module Details

We provide a detailed introduction to the four modules that make up the DiscomfortFilter.

*4.2.1 Content Filter Module.* A user can manage filtering rules directly through this module. These rules are described in natural language, specifying the discomforting recommendations the user

**Figure 4: Detailed design of DiscomfortFilter.**

wishes to avoid. Each filtering rule has an associated activation option. During the filtering phase, the module reviews each recommendation against all active filtering rules to determine if the content matches any discomforting criteria. Only recommendations that do not match any filtering rules will be shown to the user; otherwise, they will be filtered out (G4). The identification process uses a chain-of-thought (CoT) method, with the prompt detailed in Appendix C.1. Filtering records will be saved and forwarded to the Filtering Needs Discovery Module.

*4.2.2 Preference Profile Construction Module.* This module constructs a user's preference profile by analyzing the user's clicking behavior on recommendations in chronological order. This process differs from traditional personalized algorithm research in three key aspects: (1) It solely models preferences based on **individual user behavior**, rather than on the behavior of all users. (2) It offers **more comprehensive implicit feedback** by capturing both clicked items and the recommended items users choose to ignore. (3) It must be conducted in **real-time**, responding instantly to user clicks. The key to this process is summarizing the user's clicking behavior on recommended content into a preference profile made up of features and maintaining that profile over time.

As illustrated in Figure 5, we propose an LLM-based multi-agent pipeline to complete this process. We adopt the general assumption of **pairwise rank learning** [68]—when two items are displayed simultaneously, the one that is clicked is more appealing to the user. For each clicked item (denoted as *pos* for positive), the module randomly samples an unclicked item that appeared simultaneously with *pos* as a negative item (denoted as *neg*), and then constructs an ordered pair <*pos, neg*>. Note that *pos* and *neg* are unprocessed raw contents. Then, this pair is processed by the pipeline.

First, the **Perceive Agent** identifies *pos* and *neg* from the user's perspective, analyzing why the user clicked on *pos* but not on

*neg* based on the current preference profile. Different users focus on different aspects of the same item. **By incorporating a user's preference profile, the Perceive Agent can accurately identify the aspects that matter most to each individual user.**

Second, the **Summary Agent** distills the reasons for selecting *pos* over *neg* into *m pos features* and *n neg features*, drawing from the analysis of the Perceive Agent. It then forms $m \times n$ ordered pairs of <*pos feature, neg feature*> via the Cartesian product, indicating that for each pair, the user prefers the *pos feature* over the corresponding *neg feature*. **The ordered pairs that describe the partial order relationships between these features are the fundamental basis for modeling user preferences.**

Third, the **Reflect Agent** maintains a directed graph, where edges point from *neg feature* to *pos feature*, with edge weights denoting the frequency of each <*pos feature, neg feature*> pair. Upon receiving pairs from the Summary Agent, it first merges similar feature nodes and subsequently integrates them into the graph. During the merge process, candidate features for merging are first identified based on semantic similarity, and then the final merge result is obtained using LLM. **The directed graph integrates independent ordered pairs from each user click, enabling the modeling of user preferences through comprehensive structural information.**

Finally, the feature nodes are ranked using the PageRank algorithm, which **provides a comprehensive ranking of preferences**. Features with higher rankings are generally more aligned with the user's interests, while lower-ranked features tend to be less relevant.

Overall, this Module has three key characteristics: (1) Preference profile is constructed from **features** (rather than raw contents); (2) The features are summarized through **personalized content**

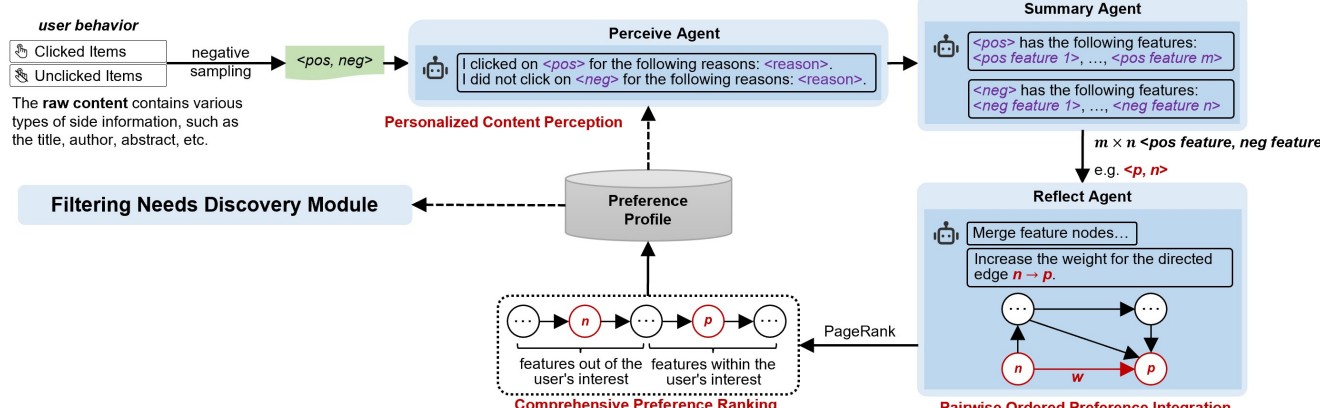

**Figure 5: The Preference Profile Construction Module is a multi-agent pipeline powered by LLMs.**

**perception**; (3) Features are **globally ranked** using the PageRank algorithm. **The preference profile constructed with these three designs helps refine the reasoning process of LLMs and significantly reduce the difficulty of aligning LLMs with user preferences**.

The preference profile is stored and sent to the Filtering Needs Discovery Module to improve the personalization of the conversational agent and provide users with clear explanations. The prompt used for this multi-agent pipeline can be found in Appendix C.2.

*4.2.3 Filtering Needs Discovery Module.* This module is a conversational agent designed to help users identify potential filtering needs with two strategies (G1). The content of each conversation round will be forwarded to the Candidate Rule Generation Module for further analysis.

**Strategy 1: Preference Profile Explanation.** This strategy begins by informing a user about the preference profile constructed by the Preference Profile Construction Module (G2). The user can then engage in multiple rounds of conversation with the conversational agent to express filtering needs, particularly where the preference profile do not match the user's expectations.

**Strategy 2: Filtering Record Explanation.** This strategy begins by informing a user about the filtering records from the Content Filter Module, including the filtered content and the reasons for filtering (G3). The user can then engage in multiple rounds of conversation with the conversational agent to refine filtering rules, particularly where the filtering records do not match the user's expectations.

It is important to emphasize that integrating the preference profile aligns the conversational agent with the user, ensuring that interactions between them are highly personalized.

*4.2.4 Candidate Rule Generation Module.* During the interaction between the conversational agent and a user, this module continuously analyzes the user's filtering needs from the conversation. Once these needs are successfully identified, the module evaluates their relevance to existing filtering rules and generates corresponding management actions. By generating actions based on relevance, the module helps prevent conflicts and redundancy. For example, if

the new filtering needs are related to existing rules, the module will generate an update action rather than create a new rule. The user can then edit and confirm the generated management actions. If no management action is confirmed (either because no needs were identified or because the user did not confirm), the conversational agent will continue interacting with the user. Figure 10 provides a detailed illustration of the above process through a flowchart.

### 4.3 Summary

**Contestability** refers to the ability to challenge decisions made by algorithms, serving as a crucial safeguard against power imbalances between users and algorithms [52]. However, this concept is often overlooked [52]. Although DiscomfortFilter does not interfere with the platform's algorithm, it fundamentally aims to **enhance user-perceived contestability in interactions with personalized algorithms**. By constructing a comprehensible and editable preference profile and allowing the user to mask any discomforting preferences, DiscomfortFilter **narrows the gap between algorithmic recommendations and user expectations**. Importantly, DiscomfortFilter inherently possesses contestability—**it does not introduce any new uncontestability while enhancing contestability in personalized algorithms**.

### 5 Evaluation

We first validate the Preference Profile Construction Module through offline experiments, then conduct a user study to assess whether the design goals help users filter discomforting recommendations and whether DiscomfortFilter meets those goals.

### 5.1 Effectiveness of Preference Profile Construction Module

*5.1.1 Task.* We validate the effectiveness of the Preference Profile Construction Module through a proxy task. Specifically, when presenting $K$ items to a user, we instruct the LLMs to predict which item the user is most likely to click based on the preference profile, with accuracy as the evaluation metric. If the preference profile is sufficiently effective, the LLMs should accurately predict the user's behavior. For each interaction sample from a user (in chronological

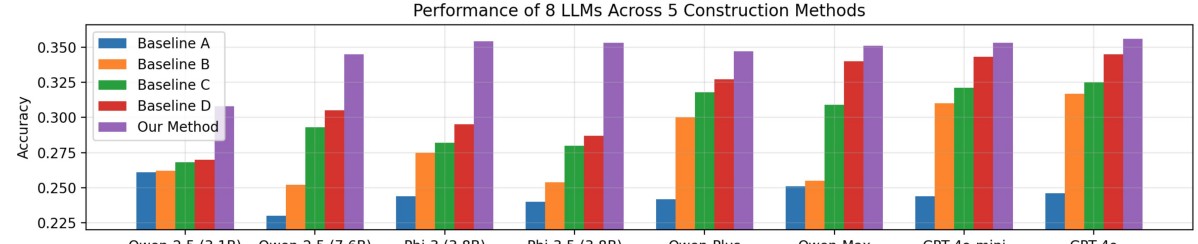

**Figure 6: Performance of 8 LLMs across 5 preference profile construction methods with $K$=4.**

**Table 1: Confusion matrix.**

|                 | Predicted Positive | Predicted Negative |
| --------------- | ------------------ | ------------------ |
| Actual Positive | 173                | 32                 |
| Actual Negative | 67                 | 208                |

order), DiscomfortFilter will initially predict the user's behavior and then observe the actual click behavior to continuously update the preference profile. Notably, DiscomfortFilter only accesses the interaction records of the individual user during this process.

*5.1.2 Dataset.* We conducted offline experiments using the MIND dataset [86], which details negative samples recommended to users during interactions, along with side information. Given the high cost of API calls, we sampled a subset of ~5,000 users for experimentation. To account for the varying interaction frequencies, we first grouped users by interaction frequency into intervals: $[0, 10)$, $[10, 20)$, and up to $[100, +\infty)$. Then, from each group, we randomly selected users, ensuring a total of at least 10,000 interactions per group.

*5.1.3 Baselines.* We conducted an ablation study comparing four heuristic in-context learning baselines for preference profile construction: (A) Preference profile exclude any user-specific information; (B) Raw contents are directly used as preference profile; (C) Features extracted from raw contents are used to construct preference profile; (D) Features are globally ranked using PageRank, but without adapting to personalized perception.

*5.1.4 Results.* We used four open-source LLMs (Qwen-2.5 (3.1B), Qwen-2.5 (7.6B), Phi-3 (3.8B), and Phi-3.5 (3.8B)) and four commercial LLMs (Qwen-Plus, Qwen-Max, GPT-4o-mini, and GPT-4o) for evaluation. Figure 6 shows their performance across various construction methods, with our method consistently achieving the best results, thus demonstrating its superior effectiveness.

To understand the reasons behind the inferior performance of other baselines, we conducted an analysis as follows: *A* fails to incorporate user-specific data, leading to essentially random predictions; *B* lacks a summary of features, and the LLMs struggle to deliver precise predictions based solely on raw content; *C* omits a summary of unclicked features, which prevents LLMs from fully modeling user preferences; *D* fails to perceive content in a personalized manner, leading to an inability to capture truly important features. **Our findings indicate that simply providing LLMs with extensive**

input is insufficient; instead, carefully curated and personalized inputs are essential for optimal performance.

The profiles constructed by several ablation baselines require sufficiently powerful LLMs to achieve better performance, leading commercial models to outperform open-source LLMs on these baseline methods. However, **our construction method reduces task complexity by streamlining the reasoning process through a well-constructed preference profile**. This enhancement enables open-source models, e.g., Phi-3, to match or even surpass the performance of certain commercial models, e.g., GPT-4o-mini.

## 5.2 User Study

*5.2.1 Implementation and Usage.* We implemented Discomfort-Filter as a third-party tool on Zhihu[3], the largest Chinese Q&A community (similar to Quora[4]). We selected Zhihu because participants frequently mentioned it during the formative study, and many participants indicated that they regularly browse personalized content on it. Furthermore, Q&A platforms are rich in user-generated content and diverse topics, and the three types of discomforting recommendations mentioned in F1 have all been observed on Zhihu.

DiscomfortFilter was implemented as a browser extension that filters discomforting recommendations by dynamically editing the DOM. We deployed Qwen2-72B-Instruct to provide LLM services for DiscomfortFilter. Aside from the LLM service, all other services run on user devices, with data stored locally. We provide three user stories and their corresponding interfaces in Appendix D to show the implementation and usage of DiscomfortFilter.

*5.2.2 Process.* We recruited **24** participants to use DiscomfortFilter freely for **one week**. Afterward, participants completed a questionnaire and randomly selected 10 filtered and 10 unfiltered items to label whether DiscomfortFilter correctly identified them. Finally, we collected usage statistics from the participants' devices and conducted **30-minute** interviews with each participant. Additional details about the process are provided in Appendix B.

*5.2.3 Preliminary Statistical Results.* On average, DiscomfortFilter processed 1,093 items per participant, filtering out 124 of them. Table 2 presents the interaction data between participants and DiscomfortFilter. The overall acceptance rate for the management actions on candidate filtering rules generated by DiscomfortFilter (via strategy 1 and strategy 2) was 93.8%.

---

[3]https://www.zhihu.com/
[4]https://www.quora.com/

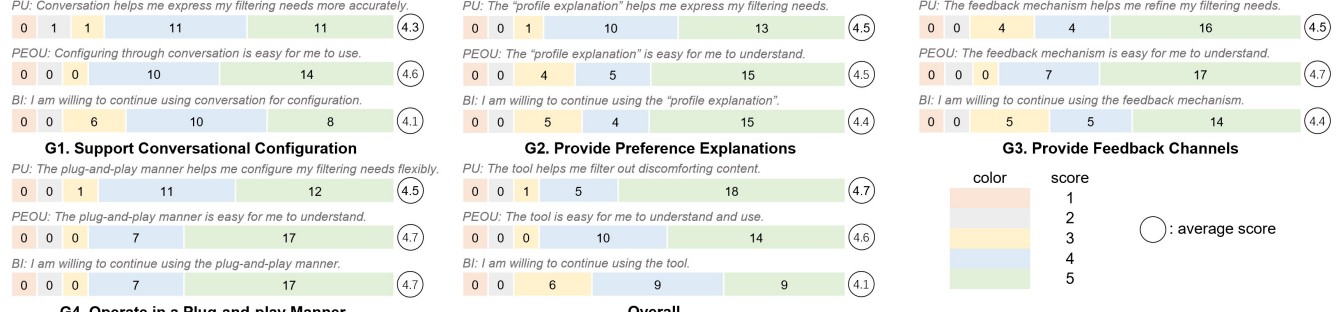

**Figure 7: Score distribution from the questionnaire: the integers in the colored bars represent the number of participants for each score, with the color-to-score relationship indicated in the bottom-right corner**

**Table 2: The statistics of participants configure the filtering rules through the Content Filter Module and the two strategies of the conversational agent.**

|  | # Messages | # Add | # Update |
|---|---|---|---|
| Content Filter Module | - | 2.9 | 2.1 |
| Strategy 1 | 13.0 | 4.4 | 2.7 |
| Strategy 2 | 22.3 | 3.2 | 7.3 |

*5.2.4 Results of the questionnaire.* Guided by the Technology Acceptance Model (TAM) [14, 55], we developed three evaluation questions for each design goal and the overall tool, focusing on perceived usefulness (PU), perceived ease of use (PEOU), and behavioral intention (BI). Figure 7 shows the specific questions and the corresponding score distributions. For each question, at least three-quarters of participants rated it 4 or 5, indicating overall satisfaction with the role of DiscomfortFilter in assisting users to filter out discomforting recommendations. With a Cronbach's $\alpha$ of 0.80, we believe the results are reliable for subsequent analysis.

*5.2.5 Results of the interview.* We compared the platform's "Not Interested" button with DiscomfortFilter and asked participants for their opinions. All participants unanimously preferred DiscomfortFilter for filtering discomforting recommendations, which can be summarized into the following five key results.

**R1. Natural language filtering rules helped participants in personalizing their filtering needs (G1).** Some participants also found emotional value in being able to "*complain to the tool about the platform's recommendations*" (P31) and "*set rules regarding mood*" (P38). However, **false associations in LLMs sometimes hindered accurate identification of discomforting recommendations.** Seven participants noted that DiscomfortFilter occasionally overextended the rules, leading to unintended content filtering. This reduced precision — as shown in Table 1, where 24 participants annotated 480 recommended contents, resulting in a precision rate of 173/240=72.1% and a recall rate of 173/(173+32)=84.4%.

**R2. Providing preference explanations helps participants identify and articulate their filtering needs (G2).** Participants were sometimes unclear about their filtering needs, and the preference explanations enabled them to "*discuss with the tool how to*

*establish rules*" (P31). However, two participants felt that **the preference profile lacked sufficient detail**, offering only a "*broad overview of the recommended content*" (P17).

**R3. Feedback channels help participants refine their filtering needs and build trust in DiscomfortFilter (G3).** Participants noted that some filtering needs were "*hard to express precisely in one attempt*" (P18). When DiscomfortFilter behaved unexpectedly, feedback channels convinced participants that "*the tool had the ability to evolve*" (P31), and encouraged them to "*refine the rules instead of abandoning the tool*" (P28).

**R4. Participants exhibited varying usage habits for the two strategies in the conversational agent (G1, G2, G3).** Participants indicated that configuring filtering rules was the most challenging aspect, and the conversational agent helped simplify this process. Most participants reported a preference for using strategy 1 to create new filtering rules because "*the gaps highlighted new filtering needs*" (P31), while strategy 2 was typically used to update existing rules due to "*errors prompting corrections to the filtering rules*" (P23). This observation is further supported by Table 2.

**R5. The plug-and-play approach facilitates flexible configuration of dynamic filtering needs (G4)**, adapting to contextual changes and short-term interests. However, three participants expressed dissatisfaction with the absence of **an efficient method for managing numerous filtering rules**, noting that "*reviewing all filtering rules to decide which to activate is cumbersome*" (P33).

*5.2.6 Case Study.* To demonstrate how participants use DiscomfortFilter, we provide an example. After searching related topics, P35 found through strategy 1 that the platform mistakenly assumed she was interested in mother-in-law and daughter-in-law relationships, which also raised privacy concerns. She then used strategy 1 to set a filtering rule: "*I do not want to see content related to mother-in-law and daughter-in-law relationships.*" Later, through strategy 2, P35 realized that DiscomfortFilter had also unintentionally filtered out content from a novel she liked (caused by false association in LLMs). She then adjusted the rule using strategy 2: "*I do not want to see content related to mother-in-law and daughter-in-law relationships, except for the fictional content in novels.*"

*5.2.7 Impact on platform recommendation outcomes.* Assuming DiscomfortFilter processes $N$ items using a filtering rule, with $n$ items identified as discomforting, we define $n/N$ as the *filtering*

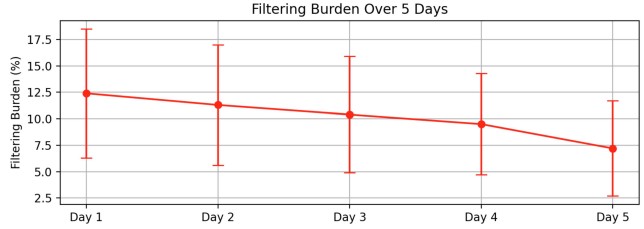

**Figure 8: Average filtering burden over 5 days.**

*burden* of this rule. We calculated the daily average filtering burden for all filtering rules that remained active for more than five days during the seven-day user study. The trend of daily average filtering burden from the day they were configured is shown in Figure 8. Over time, the filtering burden steadily declined, indicating that the platform was recommending progressively fewer discomforting items. This decline can be attributed to the introduction of DiscomfortFilter, which reduces users' exposure to discomforting recommendations, thereby resulting in fewer interactions with such items over time. As a result, the platform's recommender systems can dynamically adjust to users' evolving preferences, further reducing the likelihood of discomforting items being recommended. It is important to emphasize that, from the users' perspective, discomforting recommendations vanish immediately, as they were filtered out entirely.

## 6 Discussion

### 6.1 Challenges of Filtering Discomforting Recommendations with LLMs

Our evaluation has identified two main challenges in filtering discomforting recommendations using LLMs. **(1) False association in LLMs.** Despite careful design, LLMs sometimes misinterpret non-discomforting recommendations as containing discomforting elements, leading to unintended exclusions (R1). While LLMs' associative abilities enhance creativity, they require careful control when making decisions for users. **(2) Insufficient perceptual alignment.** Despite our meticulous design of the Preference Profile Construction Module, LLMs struggle to fully grasp users' subjective experiences, undermining the effectiveness of profile explanation (R2). A significant gap remains in helping LLMs transition from "seeing what users see" to "perceiving what users perceive".

### 6.2 Relevance to Research Topics in Recommender Systems

Recent recommender system studies increasingly emphasize user experience. **Recommendation unlearning** [48, 90], a process that allows models to forget specific user interests, enhances transparency [89] and controllability [83]. **Context-aware recommendations** [2] improve user satisfaction by accurately modeling contextual factors. **Bias-mitigating recommendations** [12] address information cocoons by prioritizing diversity [47] and fairness [84].

Our study emphasizes human-centered aspects of recommender system design, enhancing previous research primarily focused on algorithmic design. Most existing recommendation unlearning techniques [48, 90] achieve only approximate forgetting; however, integrating them with DiscomfortFilter **enables exact interest forgetting**. Traditional context-aware recommendations [2] rely solely on passively collected data, such as spatio-temporal information, while combining with DiscomfortFilter can better **meet the personalized needs of users**. Recent studies indicate that the perceived diversity among users cannot be achieved merely by providing diverse recommendations but instead relies on their active exploration [91]. While the impact of our study on information cocoons remains uncertain, we believe that, with appropriate guidance, DiscomfortFilter can enhance users' understanding of recommended content and **help them escape these cocoons actively**.

### 6.3 Potential Impact on Platforms

While DiscomfortFilter does not directly modify the platform's algorithms, it influences the items that users encounter. This influence can alter user behavior and, in turn, affect the platform's data collection and user modeling indirectly. We believe that this influence is beneficial for two primary reasons. **First**, studies indicate that even minimal control over recommendations significantly enhances users' willingness to engage [17]. DiscomfortFilter empowers users with greater control over the recommendation process, fostering trust and increasing their willingness to use the platform. **Second**, preventing data collection from users' interactions with discomforting recommendations enables the platform to model users more accurately.

### 6.4 Limitations and Future Work

We present the limitations and potential improvements from four aspects: **(1) Performance.** The two challenges outlined in Section 6.1 primarily arise from LLMs not being aligned with users. One potential solution is to introduce an interactive verification process that aligns LLMs based on user feedback. **(2) Function.** Our study currently focuses exclusively on user click behavior and text content. Future developments could incorporate other behaviors and multimodal content. Additionally, an efficient rule management solution is necessary. **(3) Evaluation.** Most participants recruited for this study hold bachelor's degrees and were assessed on a single platform over a period of one week. A large-scale deployment is needed for a long-term evaluation that encompasses a broader demographic and multiple platforms. **(4) Application.** DiscomfortFilter could be extended to other scenarios, such as parental monitoring and controlling the online content accessible to children. This requires targeted design and careful consideration of legal and ethical issues.

## 7 Conclusion

Building on insights from a formative study, we developed an LLM-based tool named DiscomfortFilter to assist users in identifying and filtering discomforting recommendations from recommender systems. Results from an offline experiment and a user study demonstrated DiscomfortFilter's effectiveness. In-depth discussions illuminated future directions and the potential broad impact of our work. We believe our study can strengthen the recommender systems community's focus on human-centered design.

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

**Table 3: Demographics of participants in the formative study. "Usage Time" refers to the total number of hours an individual spends daily on web platforms featuring personalized algorithms.**

| ID | Age | Gender | Usage Time |
|-----|-----|--------|------------|
| P01 | 23 | M | 2 |
| P02 | 27 | M | 2 |
| P03 | 23 | F | 2 |
| P04 | 24 | F | 3 |
| P05 | 23 | M | 3 |
| P06 | 30 | M | 1 |
| P07 | 24 | M | 5 |
| P08 | 22 | M | 2 |
| P09 | 23 | F | 2 |
| P10 | 26 | M | 3 |
| P11 | 24 | F | 2 |
| P12 | 24 | F | 2 |
| P13 | 51 | F | 2 |
| P14 | 35 | F | 1 |
| P15 | 45 | M | 1 |

**Table 4: Demographics of participants in the user study. "Is Zhihu User" indicates whether an individual browses personalized content on Zhihu for over 30 minutes daily.**

| ID | Age | Gender | Is Zhihu User |
|-----|-----|--------|---------------|
| P16 | 24 | M | Y |
| P17 | 24 | M | Y |
| P18 | 24 | F | Y |
| P19 | 23 | F | Y |
| P20 | 23 | M | Y |
| P21 | 23 | M | Y |
| P22 | 24 | F | Y |
| P23 | 23 | F | N |
| P24 | 22 | M | Y |
| P25 | 27 | M | Y |
| P26 | 22 | F | Y |
| P27 | 22 | F | Y |
| P28 | 25 | F | Y |
| P29 | 24 | F | N |
| P30 | 23 | M | Y |
| P31 | 24 | M | Y |
| P32 | 24 | F | Y |
| P33 | 26 | M | Y |
| P34 | 24 | F | Y |
| P35 | 35 | F | Y |
| P36 | 24 | F | Y |
| P37 | 25 | M | Y |
| P38 | 25 | M | Y |
| P39 | 27 | M | Y |

## A  The Detailed Process of the Formative Study

We recruited 15 participants (8 male, 7 female), most of whom were between the ages of 18 and 35 (13/15), representing the primary users of web platforms. They mostly spend 2-3 hours daily on various web platforms. Table 3 details the demographics of participants in the formative study.

First, we conducted semi-structured interviews to systematically understand the specific scenarios in which participants encountered discomforting recommendations on web platforms, how they handled it, and the challenges they faced. Next, we designed a draft framework and used a participatory design approach to explore their expectations for the tool. Based on their feedback, we refined our design. Each participant spent an average of about one hour on this process, and appropriate compensation was provided. All interviews were recorded with participants' consent and transcribed using automated tools as well as by the first author. To ensure privacy and security, we will not share or disclose any data.

After completing the semi-structured interviews and participatory design process, we used thematic analysis methods [56] to analyze participants' feedback and interview logs related to standard procedures [3, 58]. Initially, three authors independently conducted open coding on all feedback and collaborated thereafter to establish a set of axial codes. Subsequently, the authors reviewed interview logs, iteratively refining the coding scheme over three rounds to address any identified deficiencies. The final stage involved focused coding, aimed at synthesizing evolving conceptual categories into comprehensive topics. Throughout this process, regular communication is maintained among all authors to ensure conceptual coherence and reliability. The coding process concluded once consensus was reached among the authors regarding the conclusions.

## B  The Detailed Process of the User Study

The detailed user study process is as follows.

**Step1.** We recruited 24 participants (12 male, 12 female), all aged between 18 and 35 years. The majority of them (22/24) browse personalized content on Zhihu for over 30 minutes daily. Table 4 details the demographics of participants in the user study.

**Step2.** We first introduced the participants to the features and usage of DiscomfortFilter, and then invited them to use DiscomfortFilter at their discretion to filter out discomforting recommendations over the course of a week, without imposing any restrictions. We only intervened if participants encountered problems.

**Step3.** After using DiscomfortFilter for one week, we invited participants to complete a questionnaire to gather their feedback. The questionnaire employed a 5-point Likert scale. To assess the performance of DiscomfortFilter and its alignment with the design goals, we incorporated three dimensions into the evaluation, grounded in the Technology Acceptance Model (TAM): (1) Perceived Usefulness (PU): Assesses the effectiveness of DiscomfortFilter in helping users filter out discomforting recommendations. (2) Perceived Ease of Use (PEOU): Evaluates how easy it is for users to understand and use DiscomfortFilter. (3) Behavioral Intention (BI): Gauges users' willingness to continue using DiscomfortFilter in the future.

**Step4.** We asked participants to select 10 filtered and 10 unfiltered pieces of content and indicate whether DiscomfortFilter correctly identified each one.

**Step 5.** With participants' consent, we collected log data from their use of DiscomfortFilter and conducted interviews with each participant, averaging about half an hour in length. The interview revolved around the questionnaire, discussing DiscomfortFilter's strengths and weaknesses based on their usage experience. We then provided compensation to the participants.

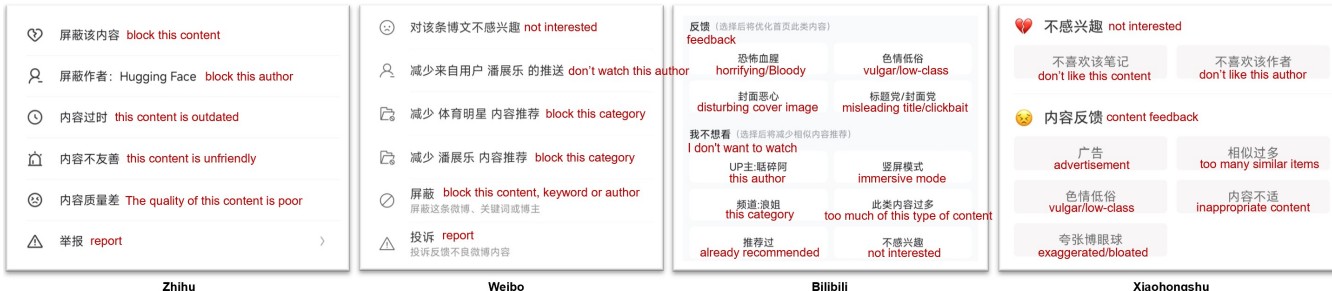

**Figure 9: The "Not Interested" button on the four most frequently mentioned social media platforms lacks personalization, flexibility, and transparency, resulting in barriers to user engagement.**

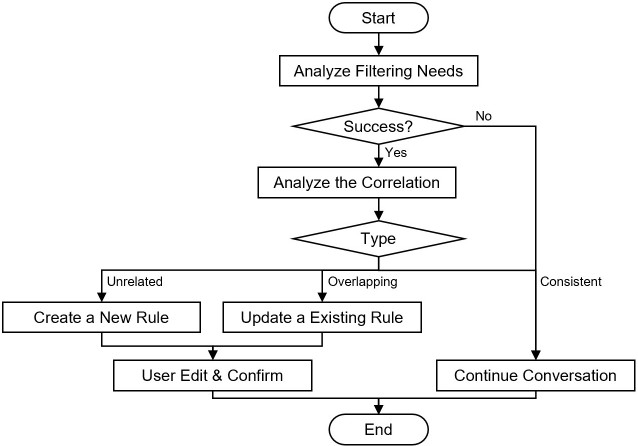

**Figure 10: The workflow of the Candidate Rule Generation Module.**

## C Prompt

The original prompt is in Chinese, but we have translated it into English for presentation purposes. All prompts include a system instruction stating, "You are a helpful assistant", which we will exclude here. In practice, we ask the LLM to provide responses in JSON format and have implemented several carefully designed constraints. However, for the sake of simplicity in presentation, we will focus only on the main logic and exclude the details.

### C.1 Content Filter Module

> **user:** There is a question on the Zhihu platform titled *<title>*, with the summary *<summary>*. Please analyze the topics that this question may relate to, ensuring not to overextend the discussion.
> **assistant:** [Response]
> **user:** A Zhihu user has established a filtering rule, *<filtering rule>*, specifying the content she/he wish to avoid. Please analyze which topics this unwanted content may relate to, ensuring not to overextend the discussion.
> **assistant:** [Response]

> **user:** Based solely on our conversation and without any additional elaboration, do you believe this user should filter out this question?
> **assistant:** [Response]

### C.2 Preference Profile Construction Module

#### C.2.1 Perceive Agent.

> Please act as a Zhihu user with the following preference information:
> Very liked: *<feature list>*
> Fairly liked: *<feature list>*
> Neutral: *<feature list>*
> Fairly disliked: *<feature list>*
> Very disliked: *<feature list>*
> Now, for a question titled *<title>*, assuming you have (or have not) interacted with it, please explain your reasons.

#### C.2.2 Summary Agent.

> A Zhihu user has (or has not) interacted with the question titled *<title>*, and here are the reasons she/he provided: *<reasons>*. Based on the reasons, please summarize the key features of the question.

#### C.2.3 Reflect Agent.

> Can *<feature>* be merged with the features listed below? If so, please provide the details of the merged feature.
> *<feature list>*

## D User Stories

Figure 11 shows the main page of DiscomfortFilter, featuring three entry points: the Content Filter Module and two conversational agent strategies. Figure 12 illustrates the workflow and corresponding interfaces for three user stories.

**Figure 11: Main page.**

**Story (a).** A patient with a mental health condition searched for related content but felt that repeated health product recommendations invaded the privacy. To address this, the user can add a rule to the Content Filter Module: "*I do not want to see content related to mental health.*"

**Story (b).** A user noticed that the platform's recommendations and her behavior showed an interest in horror content through interactions with conversational agent strategy 1. However, she doesn't want such content recommended late at night. She can express dissatisfaction and set a rule: "*I do not want to see content containing horror elements.*"

**Story (c).** Through interaction with conversational agent strategy 2, the user in story (b) noticed that some primarily comedic content with minor horror elements was mistakenly filtered. She reports this to DiscomfortFilter, which will analyze the issue and generate refined filtering rules for her to edit and confirm.

# E  Detailed Review of Related Work

## E.1  Discomforting Recommendations

Personalized algorithms infer user preferences primarily from behavioral data, often leading to an incomplete understanding of the user and resulting in discomforting recommendations [70]. Ur et al. [81] highlight that these algorithms can lead users to perceive privacy violations, particularly among individuals with psychological disorders, causing significant anxiety [22]. Moreover, these algorithms frequently fail to consider recent personal experiences, leading to inappropriate content that may trigger emotional distress [66]. The issue is compounded by the inherent popularity bias in personalized algorithms [20], which can generate discomforting recommendations for users outside the algorithm's representative groups [15, 72, 78]. This bias also increases exposure to trending misinformation [6]. Even when recommendations are accurate, users may develop negative perceptions upon realizing they are confined to a "filter bubble" [38, 69, 71]. Research on short video recommendation has identified various types of discomforting elements, including political issues, dance challenges, eating shows, and specific visual or auditory triggers [63]. Given the subjective, dynamic, and ambiguous nature of discomfort perception [65, 79], identifying such content should be guided by individual user experiences [30, 74] rather than relying solely on broad predefined categories [63].

When users are recommended discomforting content and feel misunderstood, they often attempt to influence the algorithm through their behavior [80]. However, due to their limited understanding and control over algorithmic systems [27, 28, 42, 77], users develop "folk theories"—assumptions about how the system operates [10, 19]. Based on these theories, they deliberately engage in specific behaviors to elicit desired outcomes from personalized systems. Users must be cautious in their content selection to prevent the platform from distorting their digital identity on social media [77]. When these unconventional actions fail to influence recommendations, it can lead to algorithmic irritation [24, 88], potentially triggering radical collective protests [7, 41, 54]. This frustration may further erode trust in personalized algorithms [44] and foster the feeling of being manipulated [18].

## E.2  Potential Solutions for Filtering Discomforting Recommendations

We present two categories of studies that appear relevant for identifying and filtering discomforting recommendations—one focusing on interactive recommendation systems [23, 31, 52] and the other on content moderation systems [13, 29, 36].

*E.2.1  Interactive Recommendation Systems.* An interactive recommender system utilizes data visualization techniques to create a transparent and controllable recommendation process, enabling users to actively participate in customizing personalized content. We highlighted some typical works in this field. TasteWeights [8] visualizes recommendations in three layers and allows fine-tuning of each node. Graph embedding [82] uses node-link diagrams to cluster similar movies, with filters for customization. TIGRS [9] visualizes recommendations with keyword filtering for refinement. SetFusion [64] links recommendations to techniques using a Venn diagram with color cues. PARIS [39] displays user profiles and recommendation steps, supporting user control. Bakalov et al. [5] visualize and adjust user interests using circular zones. Intent Radar [40] represents interest by keyword distance, allowing position adjustments.

*E.2.2  Content Moderation Systems.* Content moderation is used to regulate harmful content, such as hate speech and abuse [4, 21, 34, 50, 62, 87]. The process has evolved from manual review [25, 73] to automated methods using NLP technologies [11, 32, 51], which has led to reduced transparency in moderation and less feedback for users [34, 85]. To address these challenges, RECAST [85] provides mechanisms for explanation, revision, and user feedback in the moderation process. Recent research on using LLMs for content moderation [53, 57, 61, 87] highlights their potential in tasks like toxicity detection [46] and rule violation identification [43]. At the same time, there is increasing support for decentralizing content moderation, shifting control from centralized platforms to individual users, thus better enabling them to manage content they wish to avoid on social media [35]. This shift recognizes that a one-size-fits-all approach to content regulation fails to meet the diverse needs of the user base [36, 37].

## E.3  LLMs as Personal Assistants

LLMs possess semantic understanding and reasoning capabilities, enabling them to engage in human-like conversations and act as

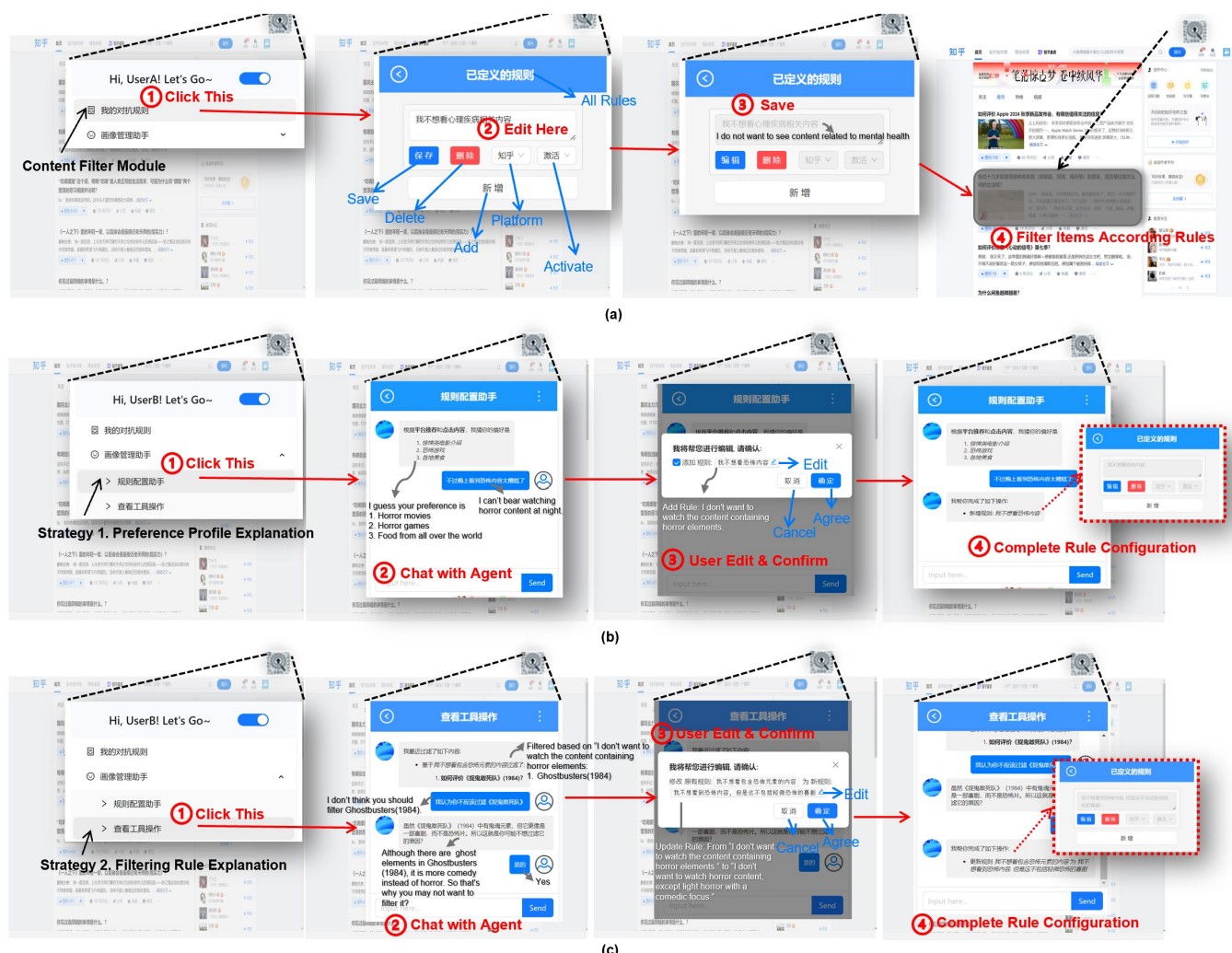

**Figure 12: The workflow and corresponding interfaces for the three user stories.**

decision-making proxies [93]. They not only help users efficiently obtain information and complete tasks but also provide more intelligent, convenient, and enriched interactions [49]. Some of them are closely integrated with personal data and devices, functioning as personal assistants [49].

Several commercial products, such as Microsoft's Copilot, New Bing, Google's Bard, and Gemini, as well as offerings from smartphone manufacturers, have integrated LLMs to boost productivity and enhance web and on-device experiences [1, 59, 60]. In the area of human-centered recommender systems, the RAH framework [76], an LLM-based multi-agent system, analyzes user preferences by observing ratings and reviews and incorporates a reflection mechanism to improve the accuracy. Recent research has explored the use of LLMs as personal assistants for various special populations. Glasser et al. [26] examined how deaf and hard-of-hearing users interact with LLM-based personal assistants capable of understanding sign language to better customize these technologies for their needs. Additionally, some work focuses on designing interfaces for visually impaired individuals to access visual information using LLMs and visual language models [92]. Finally, Jang et al. [33] investigated how autistic workers use LLMs for communication assistance, highlighting their reliance on and trust in these tools.

