# OpenReview forum: "Filtering Discomforting Recommendations with Large Language Models"
_ACM.org/TheWebConf/2025/Conference — WWW 2025 Poster_

### Official Review · Reviewer_d939 · 2024-11-30

**Novelty:** 5
**Technical Quality:** 4

**Review:**

# Strengths
- Paper is well written.
- Plug-and-play approach is interesting and provides greater transparency and controllability.
- The filtering burden shows that reduced exposure to items leads to the recommender giving fewer unwanted items over time.
- Methodology is able to personalize recommendations, even without direct access to the recommender system itself.
- Users showed high satisfaction when interacting with DiscomfortFilter.

# Weaknesses
- The number of participants in the studies is relatively small.
- In section 5.2.2, a total of 20 items are labelled. This seems like a relatively small pool of items to draw conclusions from. In addition, asking users to select the pool of items could lead to some bias. It would be better to select the pool of unseen items and ask users to select a label for each one.
- An A/B test would strengthen the argument that DiscomfortFilter is an effective tool in filtering out discomforting recommendations. For example, the user could be given the control DiscomfortFilter pipeline that lets users set filters, but the system does not actually filter out any recommendations. The user may be biased towards saying DiscomfortFilter is doing a good job because of the interface rather than actually being more satisfied that with the post-filtered results.
- No code provided for reproducibility.

**Questions:**

- Since DiscomfortFilter can effectively filter out unwanted recommendations, could a similar approach be used to instead select the best items from a set of items? The user profile is effectively capturing disinterests, but a similar method could also be used to capture interests.
- If a item is filtered out, could you also send "Not Interested" feedback to the system so that the recommender model itself has stronger feedback that the user does not want to see this item? This method may show stronger results in the filtering burden experiment in Section 5.2.7.
- Could this method be considered a DisinterestFilter rather than a DiscomfortFilter since the user is explicitly mentioning which items they are not interested in?
- Will your DiscomfortFilter code and browser extension be open-sourced?

**Reviewer Confidence:**

3: The reviewer is confident but not certain that the evaluation is correct

**Scope:**

3: The work is somewhat relevant to the Web and to the track, and is of narrow interest to a sub-community

---

### Official Review · Reviewer_Cxqb · 2024-12-02

**Novelty:** 4
**Technical Quality:** 4

**Review:**

This paper utilizes LLMs to filter out discomforting recommendations. Firstly, the paper conducts a formative study, and then, based on this study, develops an LLM-based tool named DiscomfortFilter to assist users in identifying and filtering discomforting recommendations from recommender systems.

pros:

1. The research objective is clear, the writing quality is relatively good, and a sufficient analysis of user behavior has been carried out.
2. Using LLMs as a plugin without changing the original recommendation algorithm to filter out discomforting recommendations is original, which is meaningful for the recommendation system community.
3. The authors first conduct a formative study and then conduct experiments on the Zhihu platform, verifying the effectiveness of the proposed DiscomfortFilter.

cons:

1. The offline experiment is only conducted on the MIND dataset. The number of datasets is too small and not convincing. Experiments should be carried out on at least 2 - 3 datasets.
2. In the user study part, the experimental platform and the number of users are relatively small, and most of the users have a high educational background and are not representative.
3. There is a lack of comparison with existing methods. Although there are few studies on filtering discomforting recommendations, a feasible baseline should be selected for comparison to verify the degree of improvement of the proposed method.
4. Personally, for work without open-source code, I tend to give a score one level lower. Because this non-reproducibility is not friendly to the research community.

**Questions:**

1. How does the Summary Agent obtain pos features and neg features?
2. In line 391, how is the real-time performance reflected? In the paper, I didn't see the data on the time for constructing the preference profile.

**Reviewer Confidence:**

3: The reviewer is confident but not certain that the evaluation is correct

**Scope:**

4: The work is relevant to the Web and to the track, and is of broad interest to the community

---

### Official Review · Reviewer_fLog · 2024-12-02

**Novelty:** 7
**Technical Quality:** 7

**Review:**

Summary

The paper introduces DiscomfortFilter, which is an LLM based multi-agent system that can be used to filter out recommendations to the user which can cause discomfort. This is a common problem in real world recommendation systems and the novel approach of using a plug and play add on for filtering such recs could be very relevant to several such systems.

The technique incorporates user’s preferences when coming up the filtering rules and generates a comprehensible and editable user profile that the user can modify to come up with the final filtering criteria.

The authors show that their approach works well against baselines that either are not personalized or do not use feature ranking and summarization (PageRank) across a range of LLMs. The authors demonstrate how their technique allows much smaller LLMs to match or surpass the performance of off the shelf large models such as GPT-4o.

The authors also ran a small but significant study on real users that demonstrated their filtering criteria worked well for these users. Participants in the study also mentioned how they preferred the DiscomfortFilter to the ‘NotInterested’ button that is common to most recommendation interfaces.

Strong Points
1. The paper is very well written and easy to follow.
2. Overall the paper does a good job answering most questions that one might have about the technique. The paper also does a good job outlining limitations of the current work.
3. The idea to use an additional plug and play module to filter out the recommendations is interesting and novel.
4. While the idea to use back-and-forth communication with the user to specify their likes/dislikes is not novel, the paper implements it in a nice and very user-friendly manner.
5. The idea of using a PageRank like technique to rank preferences for this task is also a novel idea.


Weak points

1. The paper mentions a few other papers that do recommendation unlearning. It would be nice to compare how well the filtering done in this paper compares with them. If this comparison is not possible, some discussion on why it is not possible would be helpful.
2. Figure 8 in the paper does a good job explaining how the feedback loop due to using the discomfort filter can help the underlying recommender model make its recommendations more relevant over time. However, it is unclear how well this effect will work for ephemeral filtering criteria.

**Questions:**

Some discussion on the weak points mentioned above would be helpful.

**Reviewer Confidence:**

3: The reviewer is confident but not certain that the evaluation is correct

**Scope:**

4: The work is relevant to the Web and to the track, and is of broad interest to the community

---

### Official Review · Reviewer_vA7n · 2024-12-05

**Novelty:** 3
**Technical Quality:** 4

**Review:**

S1: This paper is well-written and easy to follow

S2: The investigated problem of filtering discomforting items is interesting.

S3: The proposed method facilitates the discomforting recommendations filtering in a transparent and flexible way.

S4: Human studies are conducted to show the effectiveness of the proposed method.

Weaknesses:

W1. This paper only uses a QA platform for experiments, where the items in this system are basically texts. How well can this method generalize to other non-text based items?

W2: It is unclear what are the essential differences between discomforting and disliked items.

W3: The filtering process is very similar to critiquing-based recommender systems [1][2]. However, the related works are not discussed.

W4: This paper lacks a detailed discussion on the computational overhead of running the LLM-based system and how to scale up the system to real-world applications.

W5: This paper can benefit from discussing the algorithm’s effects on the echo chamber problem.

[1] Deep Language-based Critiquing for Recommender Systems

[2] Bayesian Knowledge-driven Critiquing with Indirect Evidence

**Questions:**

Please refer to W1-5

**Reviewer Confidence:**

3: The reviewer is confident but not certain that the evaluation is correct

**Scope:**

3: The work is somewhat relevant to the Web and to the track, and is of narrow interest to a sub-community